# Differential perovskite hemispherical photodetector for intelligent imaging and location tracking

Xiaopeng Feng [1], Chenglong Li[1], Jinmei Song[1], Yuhong He[1], Wei Qu [1], Weijun Li[1], Keke Guo[1], Lulu Liu[1], Bai Yang [1,2] & Haotong Wei [1,2] ✉

Advanced photodetectors with intelligent functions are expected to take an important role in future technology. However, completing complex detection tasks within a limited number of pixels is still challenging. Here, we report a differential perovskite hemispherical photodetector serving as a smart locator for intelligent imaging and location tracking. The high external quantum efficiency (~1000%) and low noise ($10^{-13}$ A Hz$^{-0.5}$) of perovskite hemispherical photodetector enable stable and large variations in signal response. Analysing the differential light response of only 8 pixels with the computer algorithm can realize the capability of colorful imaging and a computational spectral resolution of 4.7 nm in a low-cost and lensless device geometry. Through machine learning to mimic the differential current signal under different applied biases, one more dimensional detection information can be recorded, for dynamically tracking the running trajectory of an object in a three-dimensional space or two-dimensional plane with a color classification function.

Intelligent and low-cost photodetectors with advanced functions are the inevitable trend for the rapid development of modern technology[1–3]. Advances in photon detection involve diverse information such as photon efficiency[4], wide-angle vision[1,5–8], image effectiveness[1,9,10], color classification[1,11–17], object location[18,19], transmission of digital information[20], and so on. However, conventional technology in optical imaging often includes redundant, duplicate, and unrelated information, and matrix sensors in a camera also impose additional cost and complexity in imaging systems. To realize multi-function under different imaging scenes such as wide-angle, night vision, etc, conventional practice in a smartphone is to integrate several cameras, utilizing an individual different camera under different circumstances or requirements. The complex optical components and repeat pixel matrix components are actually a waste of space and cost. Fourier transform-based single-pixel imaging partially solves this issue[9,10]. A 2D image is converted into frequency domains by the Fourier transform, and only one single-pixel photodetector can record the image information by monitoring the photocurrent variations

caused by the reflected light of an object. The object image can be reconstructed according to the Fourier spectrum coefficients through inverse Fourier transform with the assistance of the Fourier algorithm.

Realizing a 2D image with a single-pixel device in a limited space is a good starting point for intelligent photodetectors, which require more and more functions to meet the requirements of modern technology. This also provides space to integrate pixel arrays with different functions, although coordination of algorithm and data processing between device pixel arrays is challenging. Machine learning has laid the foundation for intelligent technologies, which gradually innovate knowledge and modern products[1,19]. The inherent advantages of machine learning to accurately process large amounts of data enable reliable and fast development of artificial intelligence, which also provides a possibility to realize lensless color imaging with higher wavelength resolution even superior to human eyes. The advent of computational spectrometers based on computer algorithms has greatly reduced the size and cost of the spectrometer[12,17,21–24]. The basic principle to reconstruct the fine spectrum lies in the accurate

[1]State Key Laboratory of Supramolecular Structure and Materials, College of Chemistry, Jilin University, Changchun 130012, P.R. China. [2]Optical Functional Theragnostic Joint Laboratory of Medicine and Chemistry, The First Hospital of Jilin University, Changchun 130012, P.R. China. ✉e-mail: hweichem@jlu.edu.cn

relationship between the detector's response variation and the light wavelength[12,17,25]. Therefore, machine learning of the responsivity variations at different wavelengths with external bias, gate voltage, and electrode positions is expected to be able to process the data pool for high-performance computational spectrometers[17]. However, the limitation of developing intelligent photodetector through machine learning stems from the tiny variations among the pixel arrays under different circumstances.

Hemispherical surface possesses varying curvature and detection distance, which results in larger responsivity variations to the distribution of incident light intensity, wavelength, object distance information, and so on[5,18,26]. In addition, hemispherical photodetectors exhibit innate advantages of wide-angle detection in lensless systems to mimic the compound eye structure of small arthropods[5,18,19,26]. Experiments have shown that spray-coated perovskite films on a hemispherical substrate show the charge collection narrow-band (CCN) effect and wide-angle response by controlling the film composition, thickness, and charge carrier dynamics[14,26–30]. However, the photon detection mode remains rigid, which yields monotonous information, far from the ideal intelligent imaging to deliver much information.

In this article, we report a differential perovskite hemispherical photodetector with 8 differential pixels to construct an intelligent detection system. The differential pixel size, position, and responsivity under different applied biases, along with the Fourier transform algorithm and neural network fitting (NNF) assisted machine learning, enable compatibly integrate functions of interest for a wide range of applications such as computational spectrometers, wide-angle imaging, color classification, location tracking, and so on.

## Results

### Nanoparticles assisted film formation and device gain

To deposit perovskite layers on a non-flat substrate, we previously established a facile and compatible pneumatic spraying process which has been well-established in the manufacturing industry. However, the spray-coated hemispherical perovskite device exhibits a photovoltaic response with a limited external quantum efficiency (EQE) of ~10%, despite the narrow-band response in color imaging being an essential feature in intelligent imaging. To improve the device performance, we synthesized an amphiphilic molecule naphthoguanidinium iodide (NGAI) to form supramolecular aggregates, which are composed of hydrophilic guanidinium and hydrophobic naphthalene. Supramolecular aggregates are a class of materials with good dynamics, homogeneity, and accurate molecular structure[31–34]. The molecules are recrystallized in a hydroiodic acid solution to precipitate needle-like white crystals. Through X-ray single crystal diffraction, the stacked structure is obtained and is shown in Fig. 1a. The NGAI is a monoclinic crystal (P2₁/c), with the lattice parameters of a = 1.33 nm, b = 0.59 nm, c = 14.63 nm, α = γ = 90°, β = 92.27. The naphthalene groups form a layered structure driven by π-π accumulation, while several guanidinium cations are combined with iodine ions in the form of chelates (Supplementary Fig. 2d). The transmission electron microscopy (TEM) study in Fig. 1b shows that the NGAI exists in the form of a nanosheet structure. The stacking model of the nanosheet structure is similar to the single-crystal structure of NGAI. When lead iodide ($PbI_2$) is present in the solution, the morphology of the supramolecular aggregates changes into the nanoparticle structures (Fig. 1c), indicating the interaction between NGAI and $PbI_2$. The average diameter of nanoparticles is ~4.7 nm (Fig. 1c)[35]. The formation scheme is shown in Fig. 1d, $[NGA]^+$ non-polar naphthalene groups aggregate together in a polar solvent to form a dimer, and then stack laterally to form nanosheet structures, which can further evolve into nanoparticles with the presence of $PbI_2$ by forming the covered structures with $[NGA]^+$ ions.

The introduction of NGAI supramolecular aggregates is of vital importance to the device performance of differential perovskite

detectors. On one hand, the NGAI serving as an additive improves the quality and manipulates the crystalline rates of the perovskite film from spray-coated processes[36–38]. We monitored the crystallization of formamidinium lead iodide ($FAPbI_3$) perovskite precursor solutions with different equivalents of NGAI (0–30%mol relative to $Pb^{2+}$) on the substrate by time-dependent absorbance spectroscopy (700 nm, 80 °C). The addition of NGAI can effectively slow down the film crystallization rate by wrapping the $[PbI_6]^{4-}$ in solution (Figs. 1e, f)[36,39], which results in a larger grain than that film without NGAI (Supplementary Fig. 4e). On the other hand, the nanoparticles can induce the charges injection from the external circuit to output a device gain[40,41]. The EQE spectra of the photodetectors are performed to confirm the existence of device gain in Fig. 1g, and the device gain shows up with the addition of 5%mol NGAI, and the maximum EQE value of ~1000% is observed once 10%mol ~ 20%mol NGAI is employed. The device performance of differential detectors depends on their responsivity variations. The large EQE value provides a wide range of correlations between device responsivity and light signals for differential detectors to acquire more detection information. It should be also noted that the addition of NGAI promotes the crystalline orientation to the stable (111) phase as evidenced by the XRD spectra (Fig. 1h) and the morphology graph (Supplementary Fig. 4e)[42].

### Differential external quantum efficiency for computational spectrometer

The large device differential EQE/responsivity allows us to resolve the tiny difference in light wavelength number through a computer algorithm for spectrometer application. Figure 2a shows the EQE spectra to the incident light of different wavelength numbers as a function of applied reverse bias conditions. As the increase of applied reverse bias, the EQE value is greatly improved, and the short wavelength range has a larger contribution to the photocurrent, resulting in a good differential response to light wavelength. The maximum EQE value of ~1000% in Fig. 2b also yields a wide range of variations for signal simulation and machine learning to resolve two beams of light with similar wavelength numbers. Characteristic curves between the reverse bias and current density are shown in Fig. 2c under the same irradiance (50 μW cm⁻²), and the signal difference can be clearly distinguished and learned under different biases. To better understand and simulate the EQE variations versus applied reverse bias, we describe the mechanism of bias dependent charge carrier collection process in Fig. 2d. At low bias conditions, the charge carrier drift length is smaller than the film thickness, leading to a narrow-band response in EQE spectra due to the penetration depth difference between short- and long-wavelength light. Under high bias conditions, all the charge carriers can be collected regardless of where the charge is generated in the film due to the extension of charge carrier drift length across the film. Therefore, the entire range of the EQE spectra has been improved. We employed the device architecture of Cr/PTAA/Perovskites/$C_{60}$/BCP/Cr/Au, and the photodetector performance is evaluated in Supplementary Fig. 7. The current density ($J$) - voltage ($V$) curve of the photodetector exhibits a high responsivity of 5.1 A W⁻¹ under the reverse bias of -1 V and the irradiance of 10 μW cm⁻², corresponding to a high EQE value of 1180% (Supplementary Fig. 7a). The gain of the photodetector was caused by the nanoparticles induced charges injection within the perovskite film, resulting in a large specific detectivity ($D^*$) of $2 \times 10^{13}$ Jones based on the measured low noise current of ~10⁻¹³ A Hz⁻⁰·⁵ (Supplementary Fig. 7e). The response range from 350 nm to 850 nm enables the spectrometer application from UV-visible to near-infrared.

To realize the bias-dependent computational spectrometer with the differential photodetector, a computational reconstruction algorithm was applied[17]. Figure 2e shows the reconstruction spectra of four beams of monochromatic lights with the same full width at half maximum (FWHM) of ~5 nm but at different wavelengths, and the

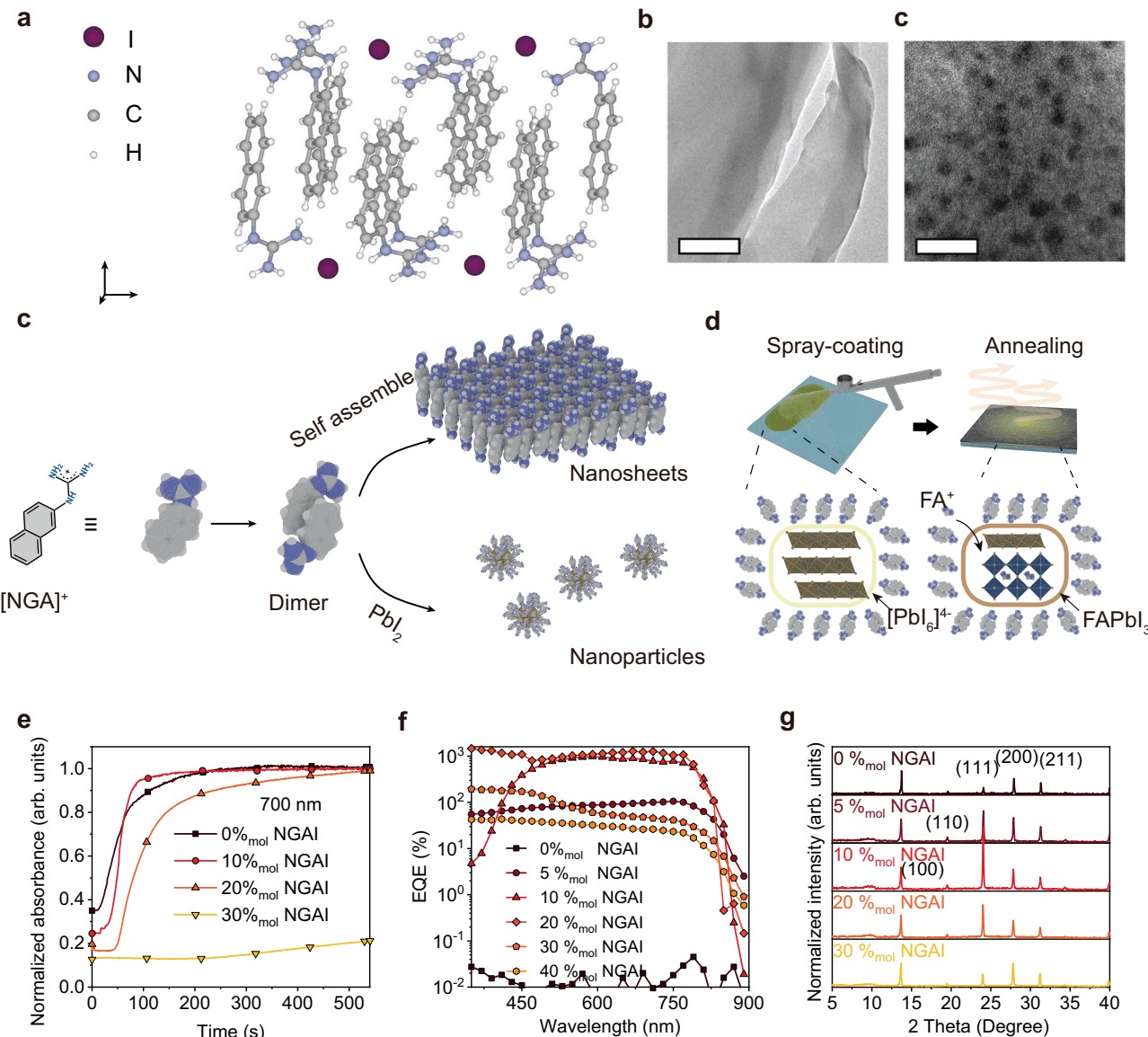

**Fig. 1 | The formation of nanoparticles and their influence on device gain. a** The crystal structure of the NGAI where NGAI is 1-(naphthalen-2-yl)guanidinium iodide. **b** The TEM image of the nanosheet structure aggregated by NGAI without adding $PbI_2$, scale bar: 200 nm. **c** The TEM image of the nanoparticles aggregated by NGAI-$PbI_2$ complex, scale bar: 20 nm. **d** Schematic diagram of the self-assembly of $[NGA]^+$ in the polar solvent (w/o and w $PbI_2$) where $[NGA]^+$ is the 1-(naphthalen-2-yl)guanidinium cation. The guanidinium of $[NGA]^+$ was exposed to the outside. **e** Schematic diagram of the process of crystallization of perovskite ($FAPbI_3$ w NGAI) during spray-coating and annealing. **f** The crystalline processes of the $FAPbI_3$ films were monitored by the absorbance at 700 nm with different amounts of NGAI under 80 °C. **g** The EQE of $FAPbI_3$ (w $0\%_{mol}$ ~ $40\%_{mol}$ NGAI) photodetectors at −1.0 V bias condition. **h** The XRD spectra of $FAPbI_3$ (w $0\%_{mol}$ ~ $30\%_{mol}$ NGAI) films fabricated by spray-coating.

simulated spectra overlap with that of incident lights. The FWHM is calculated by the reciprocal linear dispersion of the instrument, and the step size of the center wavelength is 5 nm. The corresponding resolution of the computational spectrometer was evaluated to be better than 4.7 nm in Supplementary Fig. 8a. The quasi-monochromatic light generated by a 520 nm laser with the FWHM of 5.8 nm is also reconstructed in Fig. 2f, matching well with its emission spectra. To simulate a complex situation, a polychromatic light with a white spectrum from a light-emitting diode (LED) is employed, and the differential detector can also reproduce the spectra reconstruction in Fig. 2g. Considering the influence of the irradiance, the responsivity maps with the reverse bias and wavelength are shown in Supplementary Fig. 8c–e. The reconstruction of polychromatic light with arbitrary irradiance/wavelength is feasible by building the relationship between the irradiance, wavelength, reverse bias, and current density.

## Color imaging with differential single-pixel detectors

Due to the detector's capability in color recognition, there exists significant potential in full-spectrum color imaging without the use of optical filters. This potential lies in the detector's ability to discern subtle color differences in objects. Building upon this capability, we present a design for multi-color imaging. The utilization of two-dimensional structured light patterns in conjunction with Fourier single-pixel imaging has presented a highly promising avenue for exploration. To enable single-pixel imaging with simplified device architecture and limited space, The Fourier transforms phase shift technique is adopted for high-resolution imaging[9,10]. Briefly, the set patterns were illuminated on the object through a projector, and we collected the current signal of the single-pixel device induced by the reflected light under different patterns (Fig. 3a). The collected current signal consists of the 2D image information, which can be

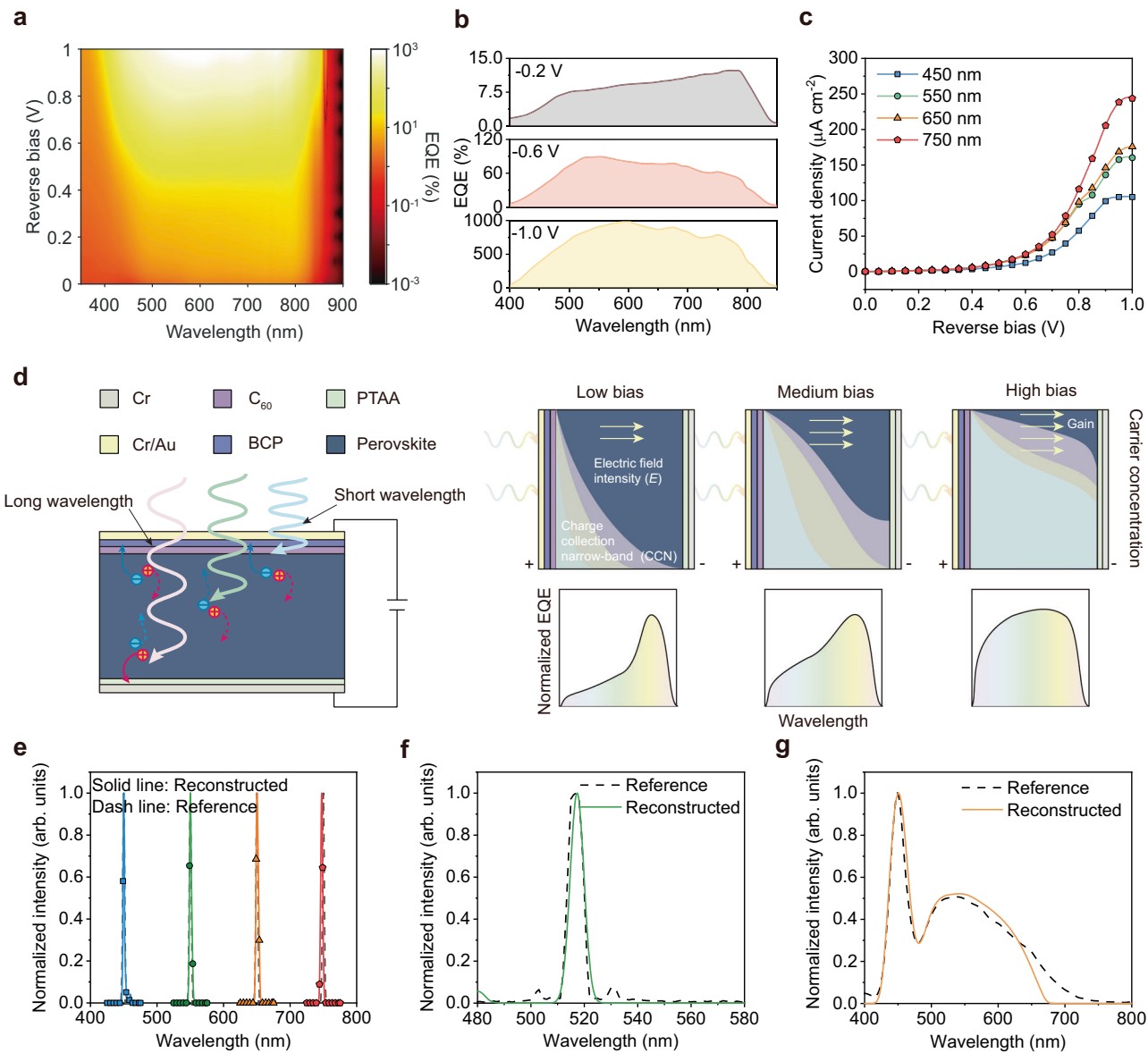

**Fig. 2 | The computational spectrometer by differential photodetectors. a** The EQE mapping of the differential photodetector at different reverse biases and wavelengths (The irradiance of light is shown in Supplementary Fig. 8b Condition 1). **b** The EQE curves of the photodetector at different reverse biases (−0.2 V, −0.6 V, −1.0 V) and wavelength. **c** The current density of the photodetector at different reverse biases and wavelengths under the constant irradiance of 50 µW cm⁻². **d** The schemed principle of the wavelength classification by differential photodetectors at different reverse biases. **e** The reconstructed spectra of four monochromatic lights by differential photodetectors match well with the reference spectra. **f** The reconstructed spectrum of the quasi-monochromatic light from a 520 nm laser. **g** The reconstructed spectrum of the polychromatic light.

reconstructed through the Fourier transform algorithm. However, the reconstructed gray 2D image doesn't include any color information. To further explore the advantages of differential detectors, bias voltage-dependent signal variations versus light wavelengths are also included in the algorithm. Since the reflected light of objects of different colors is different in the spectrum, the difference can be refined through the single-pixel imaging of objects under different reverse biases, and finally realize color imaging. In this context, we opted for a smooth Rubik's Cube as the imaging object, disregarding variations in the reflected light intensity across different positions on the Rubik's Cube's surfaces. The Rubik's cube, as shown in Fig. 3a and Supplementary Fig. 9d, was photographed by single-pixel imaging under different bias pressures (−0.1 to −0.9 V). Supplementary Fig. 9c shows the gray images of the Rubik's cube under different bias voltages. Images are optimized by reducing the noise and linear weight. The

gray-scale map of 2D images is displayed in Fig. 3b. The K-Nearest Neighbor (KNN) algorithm is used to classify the collected images (Fig. 3a and Supplementary Software 1). We select 121 pixels in the color area of each image as the training set. The gray value of a pixel at different reverse biases represents an element in an n-dimensional vector. Here, the element includes 9 dimensions referring to 9 different reverse biases. The classification accuracy obtained from the training set reaches 100%, and the confusion matrix is shown in Supplementary Fig. 10a, which yields accurate mathematical functions to classify the colors in an image in Fig. 3c. The accuracy of color classification depends on the large differential response between the responsivity and light wavelength under different reverse biases, and more refined recognition can be realized through the optimization of algorithm. While differences in reflected light intensity may pose a potential interference in color identification, the integration of

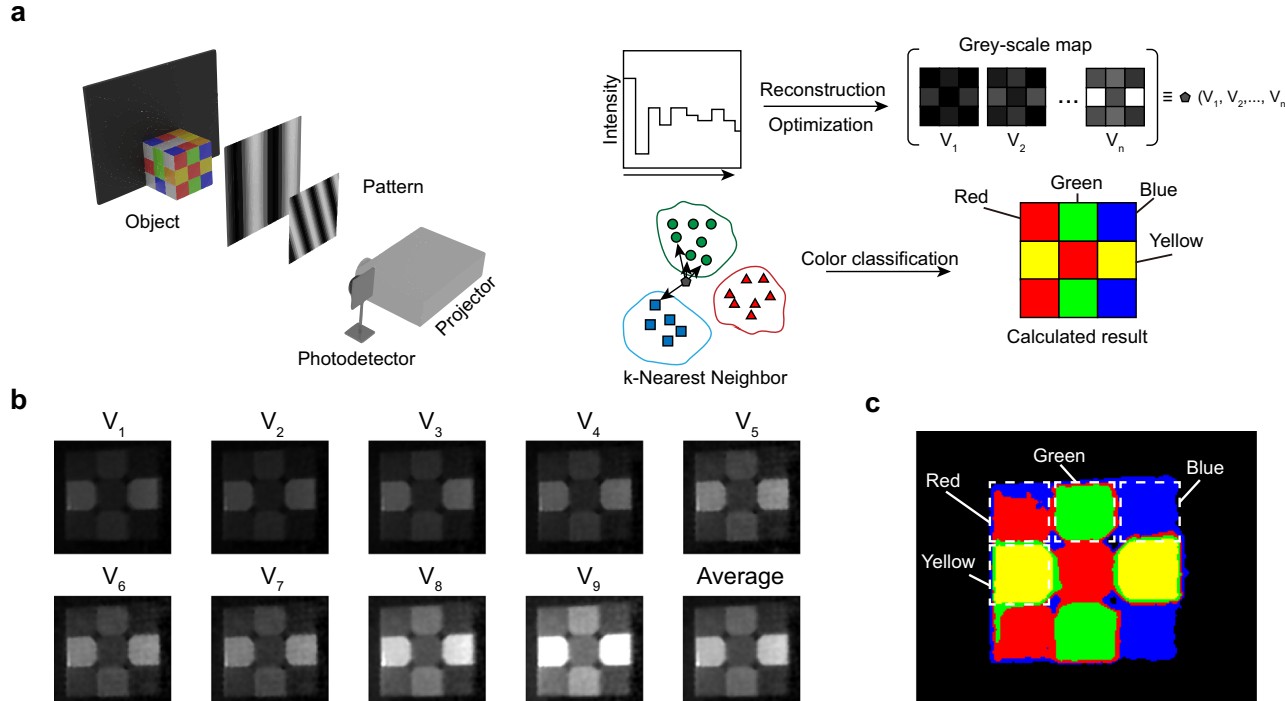

**Fig. 3 | The colorful image of the differential photodetector based on machine learning. a** Schematic diagram of the single-pixel imaging process of color classification. Patterns are projected onto the object by the projector. The reflected light is detected by the photodetector. Each pattern responded to a current density of photodetector. *m* is the numerical order of patterns. The *m*-current density curve can be translated into a gray-scale map by the algorithm. The gray level of pixels in gray-scale maps obtained under different reverse biases are collected as feature vectors. **b** The optimized images at different $V_n$ from single-pixel imaging. **c** The color image realized by a lensless differential photodetector.

spectral recognition principles and the establishment of a matrix relationship between light intensity and response hold the promise of achieving more precise color identification.

**Intelligent detection and tracking by differential detector arrays**

The spray coating method not only provides adjustable thickness and optoelectronic properties of the photodetector but also possesses the advantage of fabrication on non-planar surfaces. This feature enables the realization of a position localization system based on hemispherical detectors. To further enlarge the differential signals for intelligent detection in 3D space, we develop differential electrodes on hemispherical devices, since the effective incident flux intensity of the hemispherical surface at different positions is different compared to the planar surface[26]. Firstly, we calculate the effective incident light intensity at the surface of the hemispherical detector and planar detector depending on the different positions of the light source (Fig. 4a). *d* and *h* represent the horizontal distance and vertical distance between the light source and the devices' geometry center (hemisphere and disc), respectively. *r* is the radius of the hemisphere and disc. *l* is the distance between the light source and the device geometry center. The effective incident light intensity is uniformly distributed on the planar surface with little difference, whereas it exhibits significant variations on the hemispherical surface. Five devices with different positions (The distance from the center of the objects to the point is from 0.2 *r* to 1.0 *r*) were set on the hemispherical/planar substrates as shown in Fig. 4b to study the dependent relationship between the differential signals of the five devices as changing the horizontal distance (*d*) of the light source. The changing curves of the effective incident flux intensity at different positions of the planar surface are almost coincident. However, the differential signals at different positions of the hemispherical surface are different and asymmetric. Therefore, the differential photodetector could capture the location information of the light source.

To further realize the feature of location tracking, we designed a differential photodetector with two kinds of differential electrodes in Fig. 4c. The electrode at the bottom of the photodetector was divided into eight parts (The interval is 45°) by a mask to collect light signals from different directions as differential pixels. Two kinds of electrodes were designed to enhance the difference of the signals from different positions and improve the accuracy of the computer algorithm for location tracking. The theoretical simulation in Supplementary Fig. 12 shows the differences between the long electrode and the short electrode design in resolving the differential signals with different pixels. The hemispherical photodetector was integrated onto a printed circuit board (PCB) as a portable device prototype for a location tracker (Supplementary Fig. 13). Figure 4d shows the processes of machine learning and data reconstruction based on differential perovskite hemispherical photodetector. During the learning process, signals ($x_1$, $x_2$, ..., $x_8$) from each position of the light source were collected by every differential pixel to fit the mathematical model of Bayesian regularization (Supplementary Software 1). To simulate the location tracking function, the signals ($x'_1$, $x'_2$, ..., $x'_8$) from the light source at a series of unknown positions were collected by each differential pixel to reconstruct the position/running trajectory of the object. A LED was fixed on a precise X-Y electric multi-axis displacement table to finely control the position of the light source. Then, the differential hemispherical photodetector was placed under the light source to record the running trajectory of the LED. There are 20 steps × 20 steps in the X-Y plane with each moving step size of 3 mm to develop a signal matrix in an area of 6 × 6 cm² for 8 differential pixels. Figure 4e records the mapping results of the signal matrix by 8 differential pixels with different shapes regarding the object location tracking within a plane range, which serves as input data to build the mathematical model. The accuracy of the model generated from the neural network fitting (NNF) during the learning process can be evaluated by the characterization of the correlation coefficient in Supplementary Fig. 14. The correlation

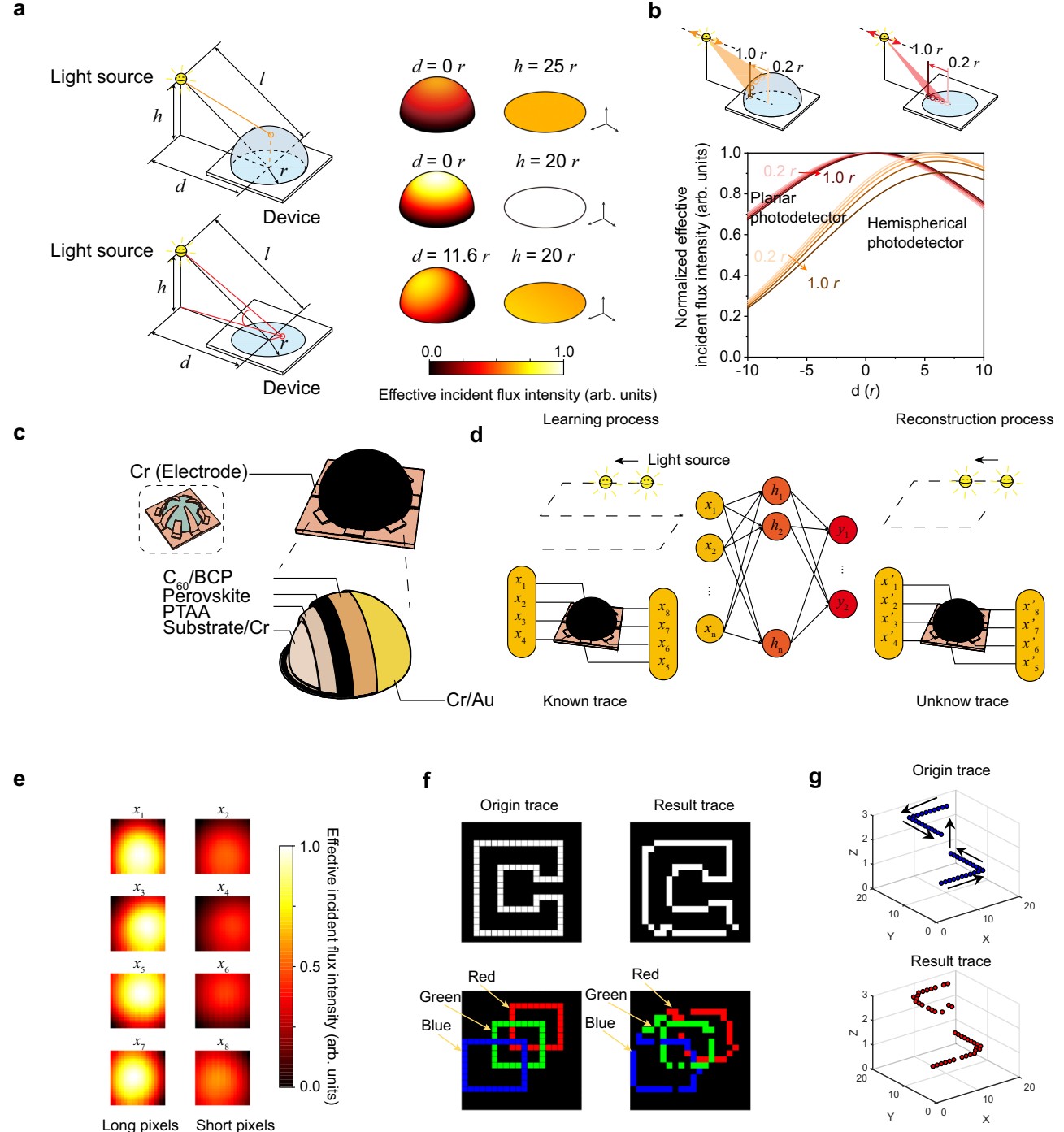

**Fig. 4 | The differential hemispherical photodetector for intelligent detection.** **a** The normalized effective incident flux on the hemispherical and planar surfaces with varying light source positions. **b** The changing curves of normalized effective incident flux at different points on the hemispherical and planar surfaces ($d = 0.2$ $r$ - $1.0$ $r$) of light at different horizontal positions. The height of the light source is constant. **c** The device structure of the hemispherical photodetector. **d** The process for building the model by NNF and trace reconstruction. **e** The signal matrix of the differential pixels distributing different positions ($x_1, x_2, ..., x_8$) of the photodetector. **f** The trace reconstruction of the light source w/o color classification and with color classification. **g** The spatial trace reconstruction in 3D space enabled by the differential hemispherical photodetector.

coefficient of the training group, test group, and all groups are 0.99976, 0.99967, and 0.99976, respectively. To verify the device performance and strategy availability, we defined the trajectory of an object in Fig. 4f (Original trace, top panel) and realized the process emersion of the motion trajectory in Fig. 4f (Result trace, top panel) and Supplementary Movie 1. The detailed trajectory reconstruction is shown in Supplementary Fig. 15.

Since all the planar location tracking was completed under a constant bias voltage, we can establish one more dimension differential signal with different biases to realize more detection information such as motion tracking with color classification feature or location tracking in 3D space. To demonstrate the color-distinguishable motion tracking, a series of signal matrixes including object color and position in the X-Y plane were built under different

biases with a correlation through NNF. Finally, the trajectories (Supplementary Movie 2) of the red, green, and blue are reconstructed with three groups of bias (−0.30 V, −0.35 V, −0.40 V), and more fine colors can be resolved if more detailed differential signals are correlated. Similarly, controlling the external bias of the photodetector can also realize location tracking in 3D space with spatial coordinates. A simple trace in a 3D array of $20 \times 20 \times 3$ was designed for proof of concept. Figure 4g (top panel) shows the original motion trace of an object in 3D space. The reconstructed trace at the bottom panel and Supplementary Movie 3 in 3D space is realized by this differential hemispherical photodetector, in consistent with the original trace.

## Discussion

In summary, we successfully proposed a differential hemispherical photodetector with spray-coated perovskite film to realize intelligent functions of color imaging, computational spectrometers, and location tracking in a 3D space or 2D plane with a color classification capability. The low noise (~$10^{-13}$ A Hz$^{-0.5}$), high EQE (~1000%), and hemispherical device architecture enable the large differential signals to collect more information of interest. Combining the advantages of differential photodetectors with machine learning with NNF processes, the most advanced photodetectors can be further enhanced. The facile design not only saves the space and cost to construct complex detector arrays, but also pushes the detector performance towards intelligence. However, the data acquisition and analysis processes still require robust computing power, which may delay the result timeliness or impair the result accuracy. Further model design and algorithm optimization are still needed to improve the maturity of differential detectors by showing advancements in intelligent performance. Through integrating differential hemispherical detector arrays, most of the advanced photodetectors can be further intelligentized and miniaturized for future artificial intelligence applications.

## Methods

### Films and devices fabrication

Films fabrication: The precursor solution based on FAPbI$_3$ perovskite was prepared in the following method. formamidine hydroiodide (FAI) 34 mg (0.2 mmol), PbI$_2$ 101 mg (0.22 mmol), methylamine hydrochloride (MACl) 4.0 mg (0.06 mmol), L-ascorbic acid (L-AA) 0.9 mg (0.025 mmol), NGAI 0–18.7 mg (0–0.06 mmol, 0–30%$_{mol}$ relative to Pb$^{2+}$) was dissolved in a mixed solvent of 1 mL DMF: 2-Me: ACN = 1: 1: 3 (v/v). The mixed solution was stirred at room temperature until completely dissolved resulting in the precursor solution for spray coating. The substrate is heated to ~100 °C, and a certain volume of solution is added to the spray gun (S-120 0.2 mm nozzle). The spray speed is 3 μL s$^{-1}$. After spraying, films were placed on a hot table at 120 °C and annealed for 15 min.

Hemispherical device fabrication: After washing with water, acetone, and isopropyl alcohol (IPA), the hemispherical substrate was deposited with chromium (Cr) electrode vacuum evaporation where the hemispherical glass was masked by patterned polyimide (PI) tape to split electrodes or define the effective area (0.1 cm$^2$). The dried substrate was treated with ultraviolet ozone for 20 min. The hemispherical substrate was fixed on the stainless steel plate and heated to 100 °C. PTAA was prepared into 0.5 mg mL$^{-1}$ toluene solution and sprayed onto the hemispherical substrate. The amount of spray solution is adjusted by referring to 25 μL cm$^{-2}$. After spraying, the Cr/PTAA substrate was thermally annealed at 100 °C for 10 min and then transferred to the plasma treatment chamber for air-plasma treatment for 2 min. The resulting film is again fixed on the stainless steel plate and heated to 100 °C. The perovskite precursor solution was sprayed onto the Cr/PTAA substrate, and after each spray deposition, nitrogen (N$_2$) was used to assist in the crystallization of the surface of the perovskite. After spraying, the device was thermally annealed at 130 °C for 15 min. The annealed device did not require any additional post-

processing and deposited the materials directly by vacuum evaporation with 25 nm C$_{60}$, 8 nm BCP, 5 nm Cr and 5 nm Au.

### Fourier single-pixel imaging and color classification

The Fourier single-pixel imaging was realized by the method published by Zhang and Mai et al.[9,10] (Additional details are presented in the Supplementary Materials.). The projector is used to project the pre-designed two-dimensional Fourier transform pattern onto the object. The detector is connected to the Keithley 2400 source meter to measure the light reflected by the object as Fig. 3a shows. The signal measured by the source meter corresponding to the first $m$ two-dimensional patterns was input to the computer ($m = 10000$), and was calculated to obtain the corresponding image through the algorithm. The image is obtained by Fourier transform of the photocurrent signal of the device under different bias voltages and shown in Supplementary Fig. 9. The images were optimized by background subtraction, noise reduction, and smoothing before linear weighting operation. The optimized images are shown in Fig. 3b with the labels V$_1$, V$_2$, ......, and V$_n$. Here, we define the RGB as different color blocks as (1, 0, 0), (0, 1, 0), (0, 0, 1), (1, 1, 0), (0, 0, 0) and the algorithm is Fine KNN. The sample size is $121 \times 5$. A certain number of pixels in the image are selected as samples, and machine learning classification is carried out through the machine learning APP in Matlab. Through the existing learning results, each pixel in the picture is classified.

### The calculation of effective incident flux intensity

The light intensity density ($I$) was defined by the following function to describe the spherical diffused light (Eq. (1)):

$$I = \frac{I_O}{4\pi R^2} \tag{1}$$

where $R$ is the distance between the object and the point light source. $I_O$ is initial light intensity. Considering the symmetry of the sphere, the effective light intensity flux distribution ($d\varphi$) can be simplified to vertical incidence as Supplementary Fig. 12 shows. The effective incident flux intensity ($d\varphi$) can be calculated in Eq. (2) by the law of cosines.

$$d\varphi = \mathbf{I} \cdot d\mathbf{S} = I \cos \beta_1 dS = \frac{I_O}{4\pi R^2} \frac{l^2 - R^2 - r^2}{2rR} dS \tag{2}$$

where $l$ is the distance between the point of the light source and the center of the sphere. $r$ is the radius of the sphere. $\beta_1$ is the angle between the incident light and the line between the point on the surface of the sphere and the center of the sphere (Supplementary Fig. 12). $S$ is the area of the surface. Similarly, $R$ can be converted into the coordinate parameter ($\theta$), which is the angle between the perpendicular line of the equatorial plane of the sphere and the line between the point on the surface of the sphere and the center of the sphere. The effective incident flux intensity can be calculated with the parameter of $\theta$ (Eq. (3)).

$$R = \sqrt{l^2 + r^2 - 2xr \cos \theta} \tag{3}$$

If the light source is placed in an arbitrary position, $l$ can be obtained by the horizontal distance ($d$) and perpendicular distance ($h$). $l$ can be calculated by Eq. (4).

$$l^2 = h^2 + d^2 \tag{4}$$

In a rectangular coordinate system, the point ($x$, $y$, $z$) on the surface of the sphere is limited by Eq. (5).

$$x^2 + y^2 + z^2 = r^2 \tag{5}$$

After the transformation of the coordinates in Supplementary Fig. 12, the distribution of effective incident flux intensity ($d\varphi$) only suffers the influence of $z$ (Eq. (6)).

$$\cos\theta = \frac{z}{r} \qquad (6)$$

In Fig. 4a, we normalized the light intensity flux distribution ($d\varphi$) in the condition of $h = 20\ r, d = 0\ r$, which is close to the experimental conditions.

The point $(x, y, z)$ at the surface of the plane is limited by Eqs. (7) and (8).

$$x^2 + y^2 \leq r^2 \qquad (7)$$

$$z = 0 \qquad (8)$$

$\beta_2$ is the angle between the line through the point $(x, y, z)$ at the surface of the plane and the position of the light source $(d, 0, h)$ and the line through the point $(x, y, z)$ at the surface of the plane and the foot point of the light source $(d, 0, 0)$ (Supplementary Fig. 12). The effective incident flux intensity ($d\varphi$) can be calculated in Eq. (9).

$$d\varphi = \mathbf{I} \cdot d\mathbf{S} = I\cos\beta_2 dS = I\sqrt{1 - \frac{h^2}{R^2}}dS \qquad (9)$$

where $R$ is the distance between the point $(x, y, z)$ and the point light source and can be calculated by Eq. (10).

$$R = \sqrt{(x - d)^2 + h^2} \qquad (10)$$

### Artificial intelligence-assisted location

The (LED 520 nm, 3 W) is fixed on the X-Y two-dimensional (2D) displacement platform, and the mobile platform is controlled by the computer to move within an area of 6 cm × 6 cm on the plane. The length of each step is 0.3 cm. The hemispherical array detector is connected in parallel to the NI9205 acquisition card. The operating voltage of the photodetector is −0.3 V. The signal of each pixel is recorded by the computer when the LED was placed in each position. The data matrix is obtained by the different values of the response of each pixel to the light signal at different positions (Fig. 4d, e). The data matrix is learned by Neural Net Fitting in Matlab, where the input is a 400 × 8 matrix representing the data: 400 samples of 8 elements. The 8 elements correspond to the signal collected by 8 pixels. The output is a 400 × 2 matrix that represents the data: 400 samples of 2 elements. The 2 elements correspond to the coordinate (X, Y) of the light source. The hidden layer contains 10 neurons, and the output layer contains 2 neurons. The fitting method is Bayesian Regularization. After that, the X-Y 2D displacement platform is used to move the LED following the prescribed route, and the signal of each detector is read by the acquisition card. The detector signal changes are brought into the NNF-fitted model for position prediction, and the corresponding coordinates are output. To realize the function of color classification, the LED was changed into 3 colors (450 nm, 520 nm, 660 nm, 3 W), and the constant reverse bias is changed into a bias group (−0.3 V, −0.35 V, −0.40 V). Thus, the data input is a 400× (8 × 3) matrix. The output is a 400 × 3 matrix where 2 elements correspond to the coordinate (X, Y) of the light source, and 1 element corresponds to the color of the LED. To realize the function of spatial orientation, the height of the LED was changed into 4 positions (9.5 cm, 9.8 cm, 10.1 cm, 10.4 cm) and the constant reverse bias is changed into a bias group (−0.3 V, −0.35 V). Thus, the data input is a 400× (8 × 2) matrix. The output is a 400 × 3

matrix where 3 elements correspond to the coordinate (X, Y, Z) of the light source.

## Data availability
The data that support the plots within this paper are available from the corresponding author upon request. The data generated in this study are provided in the Supplementary Information/Source Data file. The X-ray crystallographic coordinates for structures reported in this study have been deposited at the Cambridge Crystallographic Data Center (CCDC), under deposition numbers: 2313935. These data can be obtained free of charge from The Cambridge Crystallographic Data Center via www.ccdc.cam.ac.uk/data_request/cif. Source data are provided with this paper.

## Code availability
The source code related to the findings presented in this manuscript is available in GitHub: https://github.com/fengwind1/NCOMMS-23-43328.git.

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

## Acknowledgements

This work is financially supported by the National Natural Science Foundation of China (No. 22105083 and 52173166) and the Fundamental Research Funds for the Central Universities, JLU and JLUSTIRT (2017TD-06).

## Author contributions

H.W. conceived and supervised the project. X.F. fabricated the photodetectors and characterized the performance of the devices. C.L. helped with the photodetector fabrication and verified the device's performance. J.S. helped with the SEM measurement. Y.H. assisted in the building of the system of the read-out system of the position orientation system. W.Q. assisted with the single-pixel point imaging. W.L. contributed to the X-ray single crystal measurement. K.G. analyzed the single crystal data. L.L. performed the TEM measurement. B.Y. commented on the results and provided constructive suggestions. All authors analyzed the data. H.W. and X.F. wrote the manuscript, and all the authors commented and reviewed the manuscript.

## Competing interests

The authors declare no competing interests.

## Additional information

**Peer review information** : *Nature Communications* thanks Kai Wang, Hoon Hahn Yoon and the other anonymous reviewer(s) for their contribution to the peer review of this work. A peer review file is available.

