## [Peer Review File · Nature Communications]

Differential Perovskite Hemispherical Photodetector for Intelligent Imaging and Location TrackingREVIEWER COMMENTS

Reviewer #1 (Remarks to the Author):

This manuscript reported an intelligent detection system based on differential perovskite hemispherical photodetector arrays. By fully making use of differences of 8 pixels, such as pixel size, position, and responsivity under different applied biases, in combination with proper machine learning algorithm, the authors have realized compatibly integrate functions of interest for a wide range of applications such as computational spectrometers, wide-angle imaging, color classification, location tracking. The workload is high and the novelty can be ranked as 7/10. I would recommend “accept” so long as the following questions are well addressed:

1. Please include the reference curve when the addition of NGAI is 0% in Figure 1.g. Besides, in Fig.1 g, the color and symbols are not easy to tell. Can it be replotted?
2. The authors mention in Figure S4.e that the addition of NGAI as an additive can improve the quality of the perovskite film. However, it seems that the addition of NGAI significantly increases the gap between the perovskite grains, which could adversely affect the transport of charge carriers. Are there any methods to minimize this gap and obtain a dense perovskite film?
3. The structure of the photodetector proposed in this paper is Cr/PTAA/Perovskite/C60/BCP/Cr/Au. The Cr layer is semitransparent, which will lower the EQE. Would the author explain why they preferred Cr as the top electrode?
4. FAPbI₃ has the phase instability issue. Would the authors provides some data on stability of these devices?
5. The authors have realized different functions by different machine learning algorithm, for example, the imaging by FFT, the color by KNN and tracking by Bayesian regularization. It's interesting but looks a little messy. Is it possible to merge them into one model?

Reviewer #2 (Remarks to the Author):

The artificially intelligent sensor, so-called processing-in-sensor, is an emerging concept, inspiring many researchers in broad fields across academia and industry. The differential perovskite hemispherical photodetector reported by the authors appears to be a suitable example of a promising platform with excellent wavelength-dependent photoresponse. In this manuscript, the authors systematically studied the supramolecular aggregates of amphiphilic molecule naphthoguanidinium iodide (NGAI), significantly improving the device performance of differential perovskite hemispherical photodetector. Their efforts can be represented by high external quantum efficiency (~1000%) and low noise (10-13 A Hz^{-0.5}). In particular, the authors further developed a reconstruction algorithm by adopting neural network fitting (NNF) and successfully demonstrated color imaging and location tracking. Their findings are predictable, but demonstrations and efforts are very interesting. I think its interpretation has a good scientific value for optoelectronic scientists in the relevant community. Based on the attractive research topic and comprehensive experimental dataset, I recommend that Nature Communications accept this paper for publication after checking the following questions and suggestions.

1. It is believed that spray-coated perovskite films may be suitable for curvature surfaces such as hemispheres. However, is any data proving that applying uniform thickness to the surface is possible regardless of the curvature?

2. In Fig. 3 and Supplementary Fig. 9, the authors performed single-pixel imaging and selected 121 (11 × 11) pixels in the color area as the training set. The authors are expected to use a spatial scanning method in this process, am I right? If a spatial scanning method was carried out, the relevant description should be added to the main text, and the schematic in Fig. 3a should be updated.

3. Can the authors explain why the sudden increase in photocurrent (indicated by a red circle below) is around -0.8V in Supplementary Fig. 7a?
(image can be found in the attachment)

4. The illumination power is applied differently in three conditions, as shown in Supplementary Fig. 8. Can the author explain why the curve shape of spectral response is almost the same regardless of irradiance? Is this physically feasible?

5. The authors claim the robust stability of the FAPbI₃ (w NGAI) film fabricated by spray coating in Supplementary Fig. 6, showing the solubility and absorbance of different hydrophobic ammonium salts. However, there is no comparison of the I-V curve or detectivity between freshly prepared and aging for 30 days, like Fig. 2c and Supplementary Fig. 7a,c,f. Were these measurements repeatable in the same device and other devices? How long stable was it in the same device after fabrication for the I-V curve or detectivity? Can the authors provide an additional relevant dataset to improve the quality of the paper?

6. The demarcation energy (E_w) with the applied angular frequency ω is provided in Eq. (3), but ω_0 is not defined in the main text.

7. It would be nice to unify the y-axis scale of the reconstructed spectral curve in Fig. 2e-g and Supplementary Fig. 8a. More specifically, Fig. 2f and Fig. 2g look the same each other, but differ from Fig. 2e and Supplementary Fig. 8a.

8. The idea of differential hemispherical photodetector is great since signals are received differently and asymmetrically at each location on the hemispherical surface, so it could effectively capture the location information of the light source.

9. The bottom electrode acts as a pixel in the location-tracking read-out system. The authors used 8 pixels but claimed that more limited pixels with different shapes on the hemisphere are also possible. Is it possible to provide specific evidence or additional demonstration for this?

10. The reconstructed spectral curves presented in this paper were obtained using the algorithm provided in the previous similar study. This is an excellent example of a virtuous cycle in the research world. Likewise, I encourage the authors to open the code of their developed algorithm public for the benefit of other subsequent researchers. For example, the main algorithm of NNF is Bayesian regularization, while color classification is the K-nearest neighbor (KNN). If these two algorithm codes are open public to be archived on Github and Zenodo, they will benefit many researchers, and this paper will receive many citations after publication.

Reviewer #3 (Remarks to the Author):

In this work, Feng et al. presented hemispherical perovskite photodetectors optimized for single-pixel imaging, aiming at color differentiation and dynamic trajectory tracking. The idea of reconstructing color information through the differential EQE of the photodetector initially struck me as novel. However, the authors seemed to overlook the potential interference of light intensity in wavelength discrimination. The suggested "bias voltage-dependent signal variations relative to light wavelengths" presupposes consistent light intensity reflected from the object's surface - a scenario not very possible for 3D object imaging. Given the unique hemispherical design, light reflecting off an object would impinge on the photodetector at varied angles, resulting in inconsistent light intensity. This discrepancy complicates the task of distinguishing light colors using single-pixel imaging. It's worth noting that a hemispherical shape isn't a prerequisite for color sensing. The motion tracking potential of the hemispherical perovskite is intriguing. However, the link between this and single-pixel imaging is not explicitly delineated. A more robust evaluation, such as juxtaposing a planar device with a hemispherical lens—each equipped with machine learning for trajectory tracking—could provide a more compelling narrative. Despite presenting intriguing insights, this work lacks a cohesive narrative, rendering the ideas somewhat fragmented.

Additional comments:

The methodology behind the single-pixel FFT imaging is scarcely detailed. Given its foundational role in subsequent color differentiation, elaboration on aspects like SLM modulation and the FFT procedure is advised.

The schematic in Fig 2d, which depicts the photocarrier concentration distribution, raises questions. Under zero bias, the EQE spectrum is solely influenced by photocarriers. Introducing a bias amplifies the internal field, expediting the movement of photo-generated electrons and holes toward their respective electrodes. Yet, it's crucial to account for any charge carriers introduced to the diode under this bias—particularly when the gain exceeds one. Such carriers, devoid of light-related information, won't aid in light sensing. This could clarify the EQE's flatter curvature under substantial reverse bias. The RGB lines in Fig 2d are ambiguous. Do they represent the sum of photo-generated and injected carriers or just the former? Assuming no charge multiplication, the integrated total carrier numbers, if only photo-generated, should be the same regardless of the bias. Also, the present depiction appears speculative and demands either supplementary evidence or a detailed discussion.

Regarding motion tracking, ensuring uniformity of both the materials and device across the entire hemisphere is crucial for effective post-algorithmic data processing. It is suggested to provide further characterizations to substantiate this essential aspect.

The artificially intelligent sensor, so-called processing-in-sensor, is an emerging concept, inspiring many researchers in broad fields across academia and industry. The differential perovskite hemispherical photodetector reported by the authors appears to be a suitable example of a promising platform with excellent wavelength-dependent photoresponse. In this manuscript, the authors systematically studied the supramolecular aggregates of amphiphilic molecule naphthoguanidinium iodide (NGAI), significantly improving the device performance of differential perovskite hemispherical photodetector. Their efforts can be represented by high external quantum efficiency ($\sim 1000\%$) and low noise (10^{-13} A Hz $^{-0.5}$). In particular, the authors further developed a reconstruction algorithm by adopting neural network fitting (NNF) and successfully demonstrated color imaging and location tracking. Their findings are predictable, but demonstrations and efforts are very interesting. I think its interpretation has a good scientific value for optoelectronic scientists in the relevant community. Based on the attractive research topic and comprehensive experimental dataset, I recommend that Nature Communications accept this paper for publication after checking the following questions and suggestions.

1. It is believed that spray-coated perovskite films may be suitable for curvature surfaces such as hemispheres. However, is any data proving that applying uniform thickness to the surface is possible regardless of the curvature?

2. In Fig. 3 and Supplementary Fig. 9, the authors performed single-pixel imaging and selected 121 (11×11) pixels in the color area as the training set. The authors are expected to use a spatial scanning method in this process, am I right? If a spatial scanning method was carried out, the relevant description should be added to the main text, and the schematic in Fig. 3a should be updated.

3. Can the authors explain why the sudden increase in photocurrent (indicated by a red circle below) is around -0.8 V in Supplementary Fig. 7a?

4. The illumination power is applied differently in three conditions, as shown in Supplementary Fig. 8. Can the author explain why the curve shape of spectral response is almost the same regardless of irradiance? Is this physically feasible?

5. The authors claim the robust stability of the FAPbI₃ (w NGAI) film fabricated by spray coating in Supplementary Fig. 6, showing the solubility and absorbance of different hydrophobic ammonium salts. However, there is no comparison of the I-V curve or detectivity between freshly prepared and aging for 30 days, like Fig. 2c and Supplementary Fig. 7a,c,f. Were these measurements repeatable in the same device and other devices? How long stable was it in the same device after fabrication for the I-V curve or detectivity? Can the authors provide an additional relevant dataset to improve the quality of the paper?

6. The demarcation energy ($E\omega$) with the applied angular frequency ω is provided in Eq. (3), but ω_0 is not defined in the main text.

7. It would be nice to unify the y-axis scale of the reconstructed spectral curve in Fig. 2e-g and Supplementary Fig. 8a. More specifically, Fig. 2f and Fig. 2g look the same each other, but differ from Fig. 2e and Supplementary Fig. 8a.

8. The idea of differential hemispherical photodetector is great since signals are received differently and asymmetrically at each location on the hemispherical surface, so it could effectively capture the location information of the light source.

9. The bottom electrode acts as a pixel in the location-tracking read-out system. The authors used 8 pixels but claimed that more limited pixels with different shapes on the hemisphere are also possible. Is it possible to provide specific evidence or additional demonstration for this?

10. The reconstructed spectral curves presented in this paper were obtained using the algorithm provided in the previous similar study. This is an excellent example of a virtuous cycle in the research world. Likewise, I encourage the authors to open the code of their developed algorithm public for the benefit of other subsequent researchers. For example, the main algorithm of NNF is Bayesian regularization, while color classification is the K-nearest neighbor (KNN). If these two algorithm codes are open public to be archived on Github and Zenodo, they will benefit many researchers, and this paper will receive many citations after publication.

Response to Reviewers' Comments

Reviewer 1.

Comments 1: This manuscript reported an intelligent detection system based on differential perovskite hemispherical photodetector arrays. By fully making use of differences of 8 pixels, such as pixel size, position, and responsivity under different applied biases, in combination with proper machine learning algorithm, the authors have realized compatibly integrate functions of interest for a wide range of applications such as computational spectrometers, wide-angle imaging, color classification, location tracking. The workload is high and the novelty can be ranked as 7/10. I would recommend “accept” so long as the following questions are well addressed:

Please include the reference curve when the addition of NGAI is 0% in Figure 1.g. Besides, in Fig.1 g, the color and symbols are not easy to tell. Can it be replotted?

Response 1: We appreciate the reviewer’s recognition of our work as well as the strong suggestion of publication.

We follow the suggestion to change the symbols and the color of Fig. 1g. We also add the curve when the addition of NGAI is 0%, and the device without adding NGAI almost cannot work at -1.0 bias (EQE ~0%).

Fig. R1 The EQE of FAPbI₃ (w 0%_{mol} ~40%_{mol} NGAI) photodetectors at -1.0 V bias condition (the photodetector without adding NGAI cannot work at -1.0 bias).

Comments 2: The authors mention in Figure S4.e that the addition of NGAI as an additive can improve the quality of the perovskite film. However, it seems that the addition of NGAI significantly increases the gap between the perovskite grains, which could adversely affect the transport of charge carriers. Are there any methods to minimize this gap and obtain a dense perovskite film?

Response 2: We agree with the reviewer’s perspective that the introduction of NGAI increased the “gap” between the perovskite grains. Starting from the device’s performance, the “gaps” seem indispensable. When comparing SEM images of films with 20%_{mol} NGAI and 30%_{mol} NGAI added, the grain boundaries should be filled with the dielectric NGAI rather than left void (Materials with low electrical conductivity often appear as darker shades in SEM). Hence, the device exhibits an enhanced capability to withstand reverse bias with the leakage current suppressed. To achieve

differential incident spectroscopy, the transport of charge carriers within perovskite thin films must be regulated. However, an excessively efficient charge carrier transport capability is often disadvantageous for the spectral identification of the device. We have conducted a thorough reevaluation of relevant content within the manuscript, making modifications where descriptions were deemed inappropriate.

Comments 3: *The structure of the photodetector proposed in this paper is Cr/PTAA/Perovskite/C60/BCP/Cr/Au. The Cr layer is semitransparent, which will lower the EQE. Would the author explain why they preferred Cr as the top electrode?*

Response 3: The utilization of sputtered transparent indium tin oxide (ITO) as the top electrode holds the potential to further enhance the device performance. To save the cost and adapt to future applications, we opted for semitransparent chromium (Cr) as a substitute for ITO. Cr is characterized by commendable corrosion resistance and adhesion properties. The copious application of Cr/Au composite in electronic devices renders it a viable solution.

Comments 4: *The authors have realized different functions by different machine learning algorithm, for example, the imaging by FFT, the color by KNN and tracking by Bayesian regularization. It's interesting but looks a little messy. Is it possible to merge them into one model?*

Response 4: We appreciate the reviewer's good suggestion to establish core novelty. The core model lies in the differential device for intelligent applications. Intelligence means diverse functions, which sometimes seem messy. Real intelligent devices should be able to handle various complex and ever-changing tasks. The design of differential devices utilizes the big difference in signal response of different pixels to collect detection information of different dimensions, including imaging, color recognition, and position localization, each corresponding to distinct algorithms. The following three functions are just the potential application examples of the differential photodetectors, but are not limited to those three applications.

1. An algorithm based on the spectral reconstruction of incident light rays, which is a modification of the results from Hoon and Sun et al (*Science*, 2022, **378**, 296).
2. Correlative algorithms based on single-pixel imaging, with notable contributions made by (*Adv. Funct. Mater.*, 2021, **31**, 2104320, *Nat. Commun.*, 2015, **6**, 6225) Zhong and Mai et al. In this imaging approach, we applied a K-nearest neighbors (KNN) algorithm for color classification, primarily suited for categorical applications.
3. In the experimental reconstruction of the position of a light source using hemispherical devices, Bayesian regularization was employed to achieve the reconstruction of the light source's position. Bayesian regularization incorporates prior knowledge probabilistically to regularize models, while KNN relies on the similarity of data points in the feature space to make predictions without explicitly modeling the underlying distribution. Both methods, however, aim to improve the generalization and performance of models in different ways. Hence, in computations pertaining to

classification and fitting, there exists the theoretical possibility of algorithmic unification. The employment of diverse algorithms in this study is undertaken to harness the respective strengths inherent in each, with the overarching goal of optimizing performance.

Reviewer 2.

Comment 1: The artificially intelligent sensor, so-called processing-in-sensor, is an emerging concept, inspiring many researchers in broad fields across academia and industry. The differential perovskite hemispherical photodetector reported by the authors appears to be a suitable example of a promising platform with excellent wavelength-dependent photoresponse. In this manuscript, the authors systematically studied the supramolecular aggregates of amphiphilic molecule naphthoguanidinium iodide (NGAI), significantly improving the device performance of differential perovskite hemispherical photodetector. Their efforts can be represented by high external quantum efficiency (~1000%) and low noise (10-13 A Hz^{-0.5}). In particular, the authors further developed a reconstruction algorithm by adopting neural network fitting (NNF) and successfully demonstrated color imaging and location tracking. Their findings are predictable, but demonstrations and efforts are very interesting. I think its interpretation has a good scientific value for optoelectronic scientists in the relevant community. Based on the attractive research topic and comprehensive experimental dataset, I recommend that Nature Communications accept this paper for publication after checking the following questions and suggestions.

It is believed that spray-coated perovskite films may be suitable for curvature surfaces such as hemispheres. However, is any data proving that applying uniform thickness to the surface is possible regardless of the curvature?

Response 1: We appreciate the reviewer's recognition of our work as well as the strong suggestion of publication.

In our prior research efforts, we investigated methodologies for achieving smooth perovskite thin films through controlled layer-by-layer deposition (*Nat. Commun.*, 2022, **13**, 6106). By altering the spraying direction and iteratively accumulating thickness, we corrected and reduced the unevenness introduced by a single spraying event, ultimately obtaining a uniformly thick film. Theoretically, precise mechanical control holds the potential for the uniform deposition of hemispherical perovskite films. However, in practical experimentation, the inherent variability associated with manual spray coating introduces deviations, thereby affecting the quality of defect-free hemispherical perovskite films.

Fig. R2 The scanning electron microscopy (SEM) images of perovskite films on both planar and hemispherical substrates. **a**, High-magnification SEM image illustrating the perovskite film on the planar substrate at the cross-section. **b**, Low-magnification SEM image providing an overview of the perovskite film on the hemispherical substrate. **c**, High-magnification SEM image revealing the perovskite film on the hemispherical substrate at the cross-section. (The controlled scraping process facilitated the partial removal of surface layers in this investigation).

Comment 2: In Fig. 3 and Supplementary Fig. 9, the authors performed single-pixel imaging and selected 121 (11×11) pixels in the color area as the training set. The authors are expected to use a spatial scanning method in this process, am I right? If a spatial scanning method was carried out, the relevant description should be added to the main text, and the schematic in Fig. 3a should be updated.

Response 2: In this manuscript, the imaging methodology employed involves Fourier transform imaging using a four-step phase-shifting technique. The four-step phase-shifting technique for single-pixel imaging involves acquiring a series of four consecutive phase-shifted measurements of a target scene. These measurements are then used to calculate the phase information associated with the incident light, allowing for the reconstruction of the image through Fourier transform or similar computational methods. This approach enhances the precision of the imaging process by mitigating the impact of noise and environmental disturbances (*Adv. Funct. Mater.*, 2021, **31**, 2104320, *Nat. Commun.*, 2015, **6**, 6225). We have incorporated the specific implementation details into the supplementary information of the revised manuscript.

Comment 3: Can the authors explain why the sudden increase in photocurrent (indicated by a red circle below) is around -0.8V in Supplementary Fig. 7a? (image can be found in the attachment)

Response 3: We posit that this phenomenon may be attributed to ionization collisions, a phenomenon commonly observed in avalanche photodiodes (reach-through) (*IEEE J. Sel. Top. Quantum Electron.*, 2018, **24**, 3800208 <DOI: 10.1109/JSTQE.2017.2731938>, *Nat. Photonics*, 2020, **14**, 559). The abrupt transition in photocurrent is a prevalent occurrence within the avalanche photodiode context. This process involves the generation of electron-hole pairs due to the absorption of photons. When a reverse bias voltage is applied to the diode, the generated carriers gain energy

and create secondary carriers through impact ionization, leading to an avalanche effect (Quimby, R.S. (2006). Photodiode Detectors. In Photonics and Lasers, R.S. Quimby (Ed.). <https://doi.org/10.1002/0471791598.ch14>). The resulting multiplication of carriers significantly increases the photocurrent, making avalanche photodiodes suitable for applications requiring high sensitivity in detecting low-level optical signals. As this paper primarily focuses on the applications of the detector in color recognition and positioning, no further investigations were pursued.

APD Characteristics

Fig. R3 The $I - V$ curve of InGaAs APD (Copyright by <https://slideplayer.com/slide/7456453/>).

Comment 4: The illumination power is applied differently in three conditions, as shown in Supplementary Fig. 8. Can the author explain why the curve shape of spectral response is almost the same regardless of irradiance? Is this physically feasible?

Response 4: The curves in Supplementary Fig. 8b indicate three conditions of light intensity at different wavelength numbers, rather than the spectral response of a device. We employed neutral attenuation plates (OD 1 and OD 2) to reduce the light intensity to 1/10 and 1/100, which can be calculated according to the measured light spectra. We follow the reviewer's suggestion to modify the Supplementary Fig. 8b to avoid any confusion.

Fig. R4 The irradiance of the light source at different wavelengths.

Comment 5: The authors claim the robust stability of the FAPbI₃ (w NGAI) film fabricated by spray coating in Supplementary Fig. 6, showing the solubility and absorbance of different hydrophobic ammonium salts. However, there is no comparison of the I-V curve or detectivity between freshly prepared and aging for 30 days, like Fig. 2c and Supplementary Fig.7a,c,f. Were these measurements repeatable in the same device and other devices? How long stable was it in the same device after fabrication for the I-V curve or detectivity? Can the authors provide an additional relevant dataset to improve the quality of the paper?

Response 5: We follow the suggestion of the reviewer and provide the J - V curve of the photodetector between freshly prepared and aging for 30 days.

Fig. R5 a, The operational stability of the photodetector (R.H. 40% ~ 85%). **b,** The J - V curve of the photodetector freshly prepared and aged for 30 days.

Comment 6: The demarcation energy ($E\omega$) with the applied angular frequency ω is provided in Eq. (3), but ω_0 is not defined in the main text.

Response 6: Considering the ω_0 parameter of typical FAPbI₃ has already been established, we directly used the previously reported ω_0 from other studies (*Sci. Rep.*,

2019, **9**, 4803). We have made relevant modifications and provided citations in the supplementary materials.

Comment 7: It would be nice to unify the y-axis scale of the reconstructed spectral curve in Fig. 2e-g and Supplementary Fig. 8a. More specifically, Fig. 2f and Fig. 2g look the same each other, but differ from Fig. 2e and Supplementary Fig. 8a.

Response 7: We follow the suggestion to unify the y-axis of the Fig. 2e-g and Supplementary Fig. 8a.

Fig. R6 a, The reconstructed spectra of four monochromatic lights by differential photodetectors match well with the reference spectra. **b,** The spectra to evaluate the resolution of the computational spectrometer. Solid lines are the spectra reconstructed from the computational spectrometer. Dash lines are obtained from the monochromator.

Comment 8: The idea of differential hemispherical photodetector is great since signals are received differently and asymmetrically at each location on the hemispherical surface, so it could effectively capture the location information of the light source.

Response 8: We appreciate the reviewer's acknowledgment of our design.

Comment 9: The bottom electrode acts as a pixel in the location-tracking read-out system. The authors used 8 pixels but claimed that more limited pixels with different shapes on the hemisphere are also possible. Is it possible to provide specific evidence or additional demonstration for this?

Response 9: In the experiments, we achieved the simplified patterning of pixel points on a hemisphere by affixing patterned masks (comprising 8-pixel points) (Supplementary Fig. 13). To attain more intricate patterns, the application of reported laser scribing techniques to create finely patterned flexible masks proved viable. This will bring more intelligent functions based on our preliminary demonstration. Although there is no theoretical limitation to depositing more pixel electrodes, we are limited by the laser scribing techniques at present. Whereas it is a great direction to pursue in our future study.

Comment 10: *The reconstructed spectral curves presented in this paper were obtained using the algorithm provided in the previous similar study. This is an excellent example of a virtuous cycle in the research world. Likewise, I encourage the authors to open the code of their developed algorithm public for the benefit of other subsequent researchers. For example, the main algorithm of NNF is Bayesian regularization, while color classification is the K-nearest neighbor (KNN). If these two algorithm codes are open public to be archived on Github and Zenodo, they will benefit many researchers, and this paper will receive many citations after publication.*

Response 10: The support of previous studies has provided profound inspiration for our work. We appreciate the openness of earlier researchers in sharing their algorithmic code and adopting an open scientific attitude, which has proven beneficial to subsequent investigators. Scientific research thrives through open communication, and we acknowledge the importance of this collaborative approach. In response to the suggestions from the reviewers, we have meticulously organized the algorithm's code and publicly archived it on GitHub.

Reviewer 3.

Comment 1: In this work, Feng et al. presented hemispherical perovskite photodetectors optimized for single-pixel imaging, aiming at color differentiation and dynamic trajectory tracking. The idea of reconstructing color information through the differential EQE of the photodetector initially struck me as novel. However, the authors seemed to overlook the potential interference of light intensity in wavelength discrimination. The suggested "bias voltage-dependent signal variations relative to light wavelengths" presupposes consistent light intensity reflected from the object's surface - a scenario not very possible for 3D object imaging. Given the unique hemispherical design, light reflecting off an object would impinge on the photodetector at varied angles, resulting in inconsistent light intensity. This discrepancy complicates the task of distinguishing light colors using single-pixel imaging. It's worth noting that a hemispherical shape isn't a prerequisite for color sensing. The motion tracking potential of the hemispherical perovskite is intriguing. However, the link between this and single-pixel imaging is not explicitly delineated. A more robust evaluation, such as juxtaposing a planar device with a hemispherical lens—each equipped with machine learning for trajectory tracking—could provide a more compelling narrative. Despite presenting intriguing insights, this work lacks a cohesive narrative, rendering the ideas somewhat fragmented.

Response 1: We appreciate the reviewer's constructive suggestions on the manuscript. We also agree that the interference of light intensity has an impact on wavelength resolution, especially in a scene of active illumination. However, in real cases, most imaging employs reflected light in a passive lighting scene, which actually has little impact, since there is no such big intensity difference between reflected lights. As articulated in the manuscript, color resolution is indeed constrained by uneven light intensity. However, we believe that incorporating a sufficient amount of both light intensity and wavelength information into the model can potentially enable wavelength resolution through bias voltage adjustment. We maintain a positive outlook on this approach to color recognition, as it presents a promising avenue for achieving full-spectrum imaging through a straightforward device structure and materials.

In order to enhance the manuscript's dynamism, we conducted the following validation experiments. Initially, we measured the reflectance spectra of different facets of the Rubik's Cube under white LED illumination (Fig. R7a). The spectra were optimized using Gaussian distribution curves (Fig. R7b), and the corresponding photocurrent values were measured using a Si detector. The magnitude of the signal from the Si detector (S) can be expressed by the following equation (Eq. (3)):

$$S = \int_{w_1}^{w_2} R_{es} i dw \quad (3)$$

where w represents wavelength, w_1 and w_2 denote the lower and upper wavelength limits, respectively, R_{es} corresponds to the detector's responsivity at a specific wavelength (Fig. R7c), and i signifies the intensity of the reflected light at a particular wavelength. This approach allows for the calibration of the relative intensity of light.

Subsequently, the data previously categorized in the manuscript is multiplied by a calibration factor to approximate equalizing the light intensity. Following this step, the data is re-modeled and re-categorized, resulting in a validation of color re-classification effects (Fig. R7d). Fig. R7e shows the result after calibration and re-classification.

Fig. R7 **a**, Reflectance spectra from each face of the Rubik's cube (Background is the spectrum of the white color LED). **b**, Reflectance spectra from each face of the Rubik's cube after optimizing. **c**, The responsivity of the Si photodetector. **d**, The process of the light intensity calibration and color classification revalidation. **e**, The images after light intensity calibration and the image after color re-classification.

We have presented relevant results on trajectory tracking of planar detectors in Supplementary Table 2 and Supplementary Fig. 14c. These findings underscore the challenge of achieving positioning functionality solely through planar detectors in an array, a conclusion consistent with the outcomes depicted in Fig. 4a. In the figure, we observe that light sources from different positions result in an almost uniform distribution of effective incident flux on the plane. Consequently, there is a minimal

disparity in photocurrent among different pixel points. Therefore, establishing a positioning functionality using models based on these photocurrents proves to be challenging. In Fig. R8a, we present a schematic diagram illustrating the structure of the planar device employed for comparative analysis. Concurrently, we conducted measurements of the response matrixes corresponding to each pixel, as depicted in Fig. R8b. Notably, their responses exhibit a high degree of similarity.

Fig. R8 a, The scheme of the planar photodetector. **b**, The signal matrix of the differential pixels distributing different positions (x_1, x_2, \dots, x_8) of the planar photodetector.

We follow the reviewer's suggestion to further organize the structure of the manuscript, aiming to make it more compact and systematic.

Comment 2: Additional comments:

The methodology behind the single-pixel FFT imaging is scarcely detailed. Given its foundational role in subsequent color differentiation, elaboration on aspects like SLM modulation and the FFT procedure is advised.

The schematic in Fig 2d, which depicts the photocarrier concentration distribution, raises questions. Under zero bias, the EQE spectrum is solely influenced by photocarriers. Introducing a bias amplifies the internal field, expediting the movement of photo-generated electrons and holes toward their respective electrodes. Yet, it's crucial to account for any charge carriers introduced to the diode under this bias—particularly when the gain exceeds one. Such carriers, devoid of light-related information, won't aid in light sensing. This could clarify the EQE's flatter curvature under substantial reverse bias. The RGB lines in Fig 2d are ambiguous. Do they represent the sum of photo-generated and injected carriers or just the former? Assuming no charge multiplication, the integrated total carrier numbers, if only photo-generated, should be the same regardless of the bias. Also, the present depiction appears speculative and demands either supplementary evidence or a detailed discussion.

Regarding motion tracking, ensuring uniformity of both the materials and device across the entire hemisphere is crucial for effective post-algorithmic data processing. It is suggested to provide further characterizations to substantiate this essential aspect.

Response 2: We have incorporated the suggestions of the reviewers and provided further detailed explanations on the modulation of the Spatial Light Modulator (SLM) and the Fast Fourier Transform (FFT) procedures in the manuscript.

In the left panel of Fig. 2d, our RGB lines only represent the penetration depth of light with different wavelength numbers, which enables full-color narrow-band response by simply tuning the active layer thickness (*Nat. Photonics*, 2015, **9**, 679; *Nat. Commun.*, 2022, **13**, 6106). In contrast to the left panel of Fig. 2d, the electric field in the right panel of Fig. 2d is lateral, rather than vertical, for better comparison with the EQE spectra below. The RGB lines are employed to depict the collectible charge carriers, equaling the EQE signals. In our system, intentional modifications have been made to the defects within the perovskite film. These defects not only offer the potential for optically induced gain but also, at low bias voltages, enhance the recombination rate of charge carriers within the perovskite film. This facilitates a more pronounced current response for carriers generated near the absorption edge, known as the charge collection narrowing effect (CCN) or narrow-band response. Consequently, under different reverse biases, the extractable charge carriers are wavelength-dependent. At higher biases, the device's response is the gain favored. To avoid any confusion, we modify Fig. 2d by adding electrodes, and more discussions are also added (Fig. R9).

Fig. R9 The schemed principle of the wavelength classification by differential photodetectors at different reverse biases.

We follow the reviewer's suggestion to conduct an assessment of the uniformity of the hemisphere film. The spray-deposited perovskite film exhibits uniformity across the entire hemisphere. Fig. R10b presents cross-section SEM images obtained by selectively scratching portions of the perovskite film from the hemisphere. Additionally, experimental results from the operation of the hemisphere photodetector affirm that the obtained uniformity of the perovskite film does not significantly impact the device's performance, rendering it acceptable.

Fig. R10 a, Low-magnification SEM image providing an overview of the perovskite film on the hemispherical substrate. **b**, High-magnification cross-section SEM image revealing the perovskite film on the hemispherical substrate. (The scratching process leads to the partial removal of surface layers in this investigation).

REVIEWERS' COMMENTS

Reviewer #1 (Remarks to the Author):

Thank the authors. I have no more questions.

Reviewer #2 (Remarks to the Author):

I have received more than adequate responses to all of the comments I have made. There are a few minor reverse questions, but I guess there is no need to delay the publication of this paper. Thus, I recommend acceptance of this paper for publication.

One last thing: Following my comment #10, the authors mentioned they publicly archived their algorithm's codes on GitHub, but I cannot find information or links. Thus, I would appreciate it if the editor could ask them again to provide GitHub links (written in the manuscript or supplementary information) both for the trajectory tracking algorithm (neural network fitting by Bayesian regularization) and color classification algorithm (K-nearest neighbor method), respectively.

Reviewer #3 (Remarks to the Author):

My questions have been well addressed. I do not have further concerns about this work.

Response to Reviewers' Comments

Reviewer 1.

Comment 1: Thank the authors. I have no more questions.

Response 1: We also appreciate the valuable suggestions from the reviewers, whose contributions significantly enhanced the publication of the article.

Reviewer 2.

Comment 1: I have received more than adequate responses to all of the comments I have made. There are a few minor reverse questions, but I guess there is no need to delay the publication of this paper. Thus, I recommend acceptance of this paper for publication.

One last thing: Following my comment #10, the authors mentioned they publicly archived their algorithm's codes on GitHub, but I cannot find information or links. Thus, I would appreciate it if the editor could ask them again to provide GitHub links (written in the manuscript or supplementary information) both for the trajectory tracking algorithm (neural network fitting by Bayesian regularization) and color classification algorithm (K-nearest neighbor method), respectively.

Response 1: We have incorporated the suggestions from the reviewers and have subsequently organized the code. The refined code has been uploaded to GitHub at the following repository: <https://github.com/fengwind1/NCOMMS-23-43328>. Please feel free to reach out to us if there are any further inquiries or issues.

Reviewer 3.

Comment 1: My questions have been well addressed. I do not have further concerns about this work.

Response 2: We express our gratitude for the valuable suggestions and questions raised by the reviewers during the process. The publication of this work owes much to the diligent efforts of the reviewers.